# Improving Code Localization with Repository Memory

**Boshi Wang**[♠][*]    **Weijian Xu**[◊]    **Yunsheng Li**[◊]    **Mei Gao**[◊]    **Yujia Xie**[◊]
**Huan Sun**[♠]    **Dongdong Chen**[◊]

[♠]The Ohio State University    [◊]Microsoft
`wang.13930@osu.edu`

## Abstract

Code localization is a fundamental challenge in repository-level software engineering tasks such as bug fixing. While existing methods equip language agents with comprehensive tools/interfaces to fetch information from the repository, they overlook the critical aspect of *memory*, where each instance is typically handled from scratch assuming no prior repository knowledge. In contrast, human developers naturally build long-term repository memory, such as the functionality of key modules and associations between various bug types and their likely fix locations. In this work, we augment language agents with such memory by leveraging a repository's *commit history*—a rich yet underutilized resource that chronicles the codebase's evolution. We introduce tools that allow the agent to retrieve from a non-parametric memory encompassing recent historical commits and linked issues, as well as functionality summaries of actively evolving parts of the codebase identified via commit patterns. We demonstrate that augmenting such a memory can significantly improve LocAgent, a state-of-the-art localization framework, on both SWE-bench-verified and the more recent SWE-bench-live benchmarks. Our research contributes towards developing agents that can accumulate and leverage past experience for long-horizon tasks, more closely emulating the expertise of human developers.

## 1 Introduction

Repository-level software engineering tasks, such as bug fixing, are a promising application for Large Language Model (LLM)-powered agents (Jimenez et al., 2024). A crucial first step in these tasks is **code localization**: identifying the specific files and code segments that need to be modified to resolve the issue at hand. Existing methods mainly focus on building powerful toolsets that help agents navigate and reason over code relationships (Liu et al., 2025; Yu et al., 2025; Ouyang et al., 2025; Chen et al., 2025b; Ma et al., 2025). A leading framework is LocAgent (Chen et al., 2025b), which parses codebases into directed heterogeneous graphs that capture code structures and dependencies, enabling effective search for relevant entities.

Despite steady progress, current approaches share a key limitation: they treat every problem as a fresh puzzle, solved from scratch assuming no prior knowledge of the repository. Human developers, by contrast, accumulate and leverage long-term repository memory over time—this includes cached understanding of the purpose of core and actively evolving modules, and various associations between recurring bug patterns and their likely fix locations. This accumulated memory is what allows developers to grow into experts in a codebase.

The importance of such memory is also clear when looking at failure cases of existing localization frameworks. To illustrate, consider a failure case of LocAgent on a bug in the `django` repository from SWE-bench, as illustrated in Figure 2. The example is about `django`'s migration system, which generates migration programs from a user-defined schema. Here, the challenge is to find where import statements for certain base classes are synthesized, since the bug stems from missing imports in the generated program. Without prior knowledge of the repository, an agent must embark on a

---

[*]Work done as an intern at Microsoft.

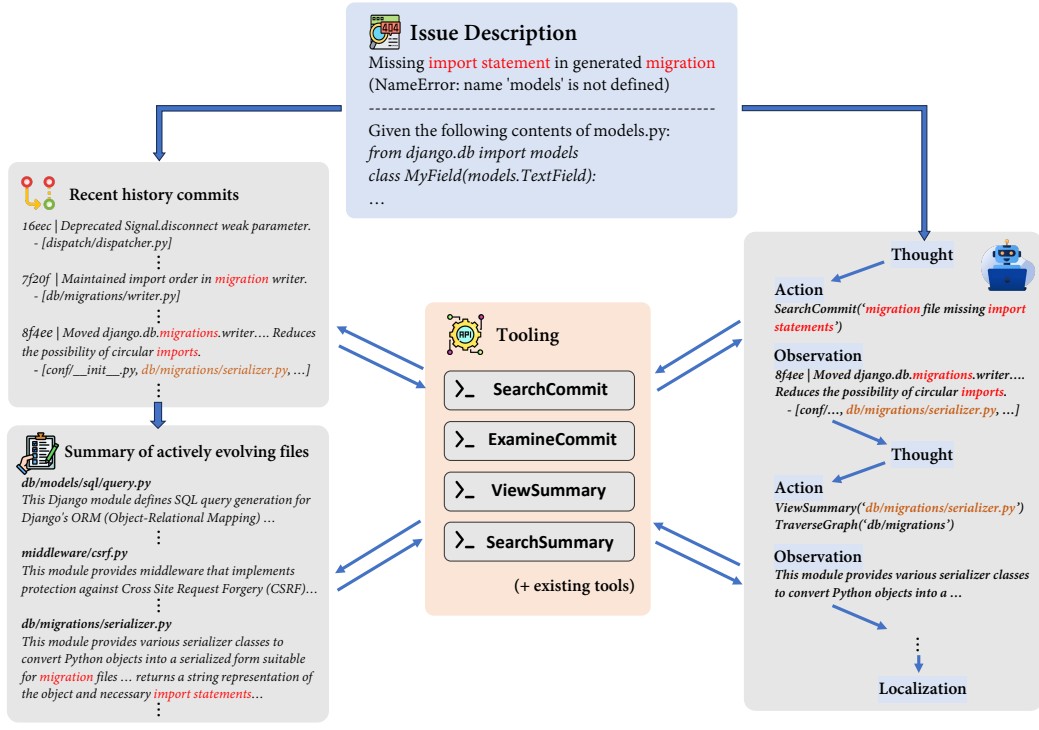

Figure 1: An overview of our repository memory design. **(a)** We construct the memory by leveraging the recent commit history of the repository. This involves creating a searchable database of past commits and their linked issues, and identifying frequently edited files to let LLMs generate high-level functionality summaries. **(b)** The memory is accessed by the language agent via a set of tools that perform search based on custom queries and support closer examination of individual memory entries. Details in §3.

complex investigation, carefully tracing data/control flows and function calls across different folders and files to find the source of the error. In this example, while the agent successfully located some initial key entities, it eventually failed to complete the reasoning chain and stopped prematurely.

Experienced developers would likely approach the problem differently. They could draw on *episodic memory* of past issues/codebase changes related to the migration system, or recall from *semantic memory* the modules that are potentially responsible for handling such imports within the codebase. These memories could provide strong priors for the investigation, guiding the search/reasoning to more effectively reach the error source.

How can we equip agents with such kind of memory? We propose to leverage the repository's *commit history*—a natural record of its past evolution. In particular, new problems are often connected to some past changes, where the related commit patches and linked issue contents could provide valuable data source for approximating the episodic memory. Commit statistics could also naturally reveal which parts of the codebase are most active, making them prime candidates for building semantic memory. Returning to the example in Figure 2, we find that even a simple keyword search (*"migration"*, *"import"*) over the commit messages retrieves many related history patches in the `django` migration system, such as problems with circular imports and nested classes. Likewise, analyzing commit frequency highlights the target file as a module under active development, and a pre-computed summary of its functionality—managing object-to-string conversion and import statements—could provide a strong signal of its relevance to the issue.

Building on these intuitions, we design two simple memory mechanisms to augment existing frameworks:

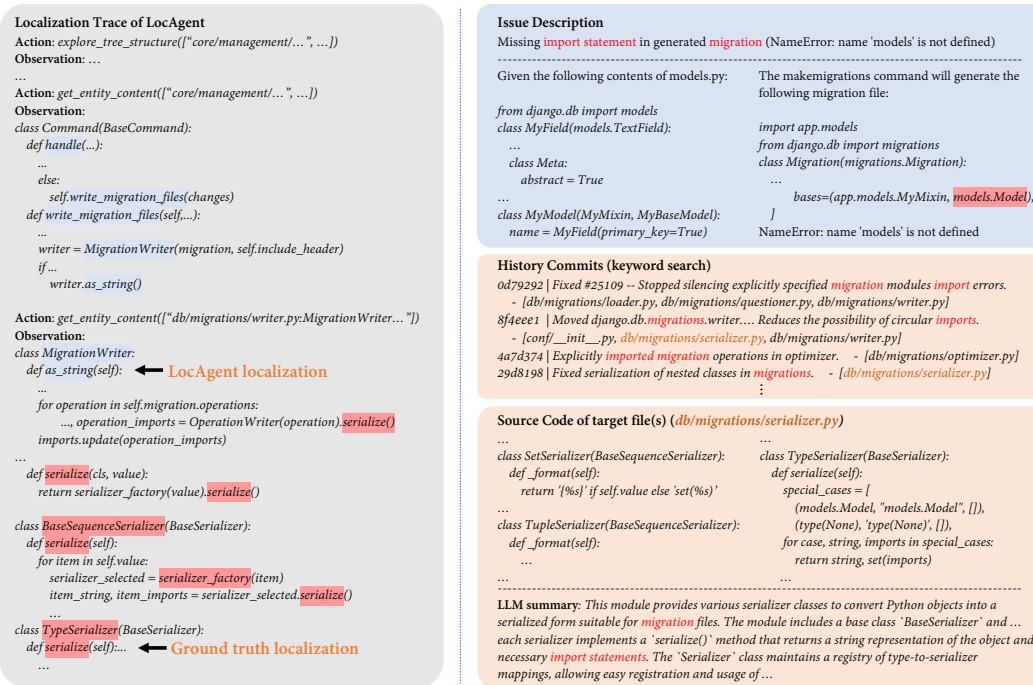

Figure 2: **(Left)** Localization trajectory of a failure case of LocAgent on SWE-bench-verified (*django__django-14580*). While the agent successfully traces some initial key entities, it fails to reason in greater depth and granularity to pinpoint the error source, resulting in wrong localizations. **(Right)** The original issue description (top), accompanying history commits obtained via simple keyword search on commit messages (middle), the source code and LLM-generated functionality summary of the ground truth target file containing the error source (bottom).

- **Episodic Memory of Past Commits.** We crawl and preprocess the commit history and linked data, and provide tools for agents to 1) search this corpus via custom queries that are matched with the commit messages, and 2) examine the details of individual commits, such as linked issues and commit patches. The episodic memory allows agents to reference past codebase changes to aid in resolving the current issue.

- **Semantic Memory of Active Code Functionality.** We identify the most active parts of the codebase by analyzing commit frequency to find the most frequently edited files. For these key files, we use an LLM to generate high-level summaries of their functionalities. This creates a compact knowledge base of the most dynamic parts of the codebase, which the agent can query to understand the purpose of potentially relevant modules.

Experiments show that augmenting LocAgent with these memory components could significantly improve localization performance, where we observe strong gains on both SWE-bench-verified (Jimenez et al., 2024) and the more recent SWE-bench-live (Zhang et al., 2025) benchmark.

To summarize, we make the following contributions:

- We identify the lack of long-term memory as a critical limitation in current language agents for repository-level software engineering tasks such as code localization.

- We propose to leverage commit history as a natural and rich source for building repository memory, and introduce two simple memory mechanisms—episodic memory of past commits and semantic memory of active code functionality—that integrate easily into existing frameworks.

- We show that these mechanisms yield substantial improvements in code localization, highlighting the value of incorporating long-term memory into agent workflows.

## 2 RELATED WORK

### 2.1 CODE LOCALIZATION IN REPOSITORY-LEVEL SOFTWARE ENGINEERING TASKS

Existing methods for code localization could be broadly categorized into three types: 1) retrieval-based, 2) agentic approaches and 3) procedural approaches.

**Retrieval-based methods** represent the most conventional approach, leveraging lexical or semantic matching to rank code snippets based on their proximity to the query (Wang et al., 2025b). Recent advances have focused on improving the quality of code embeddings, often through contrastive learning objectives (Li et al., 2022; Wang et al., 2023; Zhang et al., 2024a; Suresh et al., 2025). Among these, CoRNStack (Suresh et al., 2025) represents the current state of the art, achieving strong performance through large-scale training coupled with rigorous data filtering and hard negative mining strategies.

**Agentic frameworks.** Agentic approaches augment LLMs with the ability to interact with an external environment to gather information via a set of tools/interfaces, where the major focus has been to improve the comprehensiveness of the tool designs (Yang et al., 2024; Cognition, 2024; Örwall, 2024; Zhang et al., 2024b; Chen et al., 2025b; Wang et al., 2025a; Yu et al., 2025; Ouyang et al., 2025; Ma et al., 2025; Liu et al., 2025). Notably, LocAgent (Chen et al., 2025b) is a SoTA agentic framework for code localization. It parses codebases into heterogeneous graphs capturing code structures and various kinds of dependencies (e.g., import, invoke and inherit relationships), which allows LLM agents to more effectively comprehend and navigate through the codebase.

**Procedural approaches** directly employ LLMs to perform localization in a pre-designed procedure (Zhang et al., 2023; Wu et al., 2024; Xia et al., 2025; Liang et al., 2024), which avoids the complex setups of agentic approaches. The most representative and high-performing method is Agentless (Xia et al., 2025), which performs localization by prompting LLMs with the issue description and a concise representation of the repository's file and directory structure.

### 2.2 JUST-IN-TIME SOFTWARE DEFECT PREDICTION

Our work shares basic intuitions with the field of Just-in-Time Software Defect Prediction (JiT-SDP), which aims to predict the likelihood that a specific code change (e.g., commit) is defective immediately after submission (Zhao et al., 2023). JiT-SDP leverages historical data—such as code churn, diffusion, developer experience, and commit message metadata—to assess risk (Kamei et al., 2012; Yang et al., 2015). While JiT-SDP focuses on *predicting* the introduction of bugs based on evolutionary patterns, our framework utilizes similar historical signals (commit history and file activity) to help agents *localize* and fix reported bugs. Specifically, we adopt the intuition from JiT-SDP that "hotspots" (frequently modified files) and historical commit contexts are strong indicators of where defects and their solutions reside.

### 2.3 MEMORY-ENHANCED LANGUAGE AGENTS

Our work is connected with the broader literature on enhancing language agents with memory or experience (Qian et al., 2024; Chen et al., 2025a; Wang et al., 2024; 2025c; Zheng et al., 2025). The most related work is arguably Chen et al. (2025a), which distills procedural knowledge from an agent's past success and failure trajectories to facilitate online problem-solving. Orthogonally, our approach leverages commit histories to construct a repository-specific memory, providing knowledge that is grounded in the codebase's evolution rather than the agent's individual experience.

## 3 REPOSITORY MEMORY

To bridge the gap between memoryless agents and experienced developers, we tap into the repository's commit history—a rich, structured chronicle of its evolution. We structure this historical data into two complementary memory stores, designed to be lightweight and easily integrated into existing agentic frameworks. The first, an episodic memory, captures the narrative of specific past changes. The second, a semantic memory, distills high-level functional knowledge about the codebase's most dynamic areas. An illustration of this design is provided in Figure 1.

## 3.1 Episodic Memory of Past Commits

**Memory Construction.** This memory captures concrete entries of past problems and their solutions. We build a structured corpus from the repository's recent commit history, storing the code patches and also the rich metadata surrounding them: commit messages, timestamps, and links to associated issues. The corpus only includes commits made prior to the issue to be resolved (to avoid contamination). We further filter this datastore to remove issues that have overlapping text with the test instance and commits that are linked to these issues, to prevent leakage.

**Memory Interfaces.** The agent interacts with this historical database through a dedicated interface, allowing it to query for past events that are related to its current task:

- `SearchCommit(query, top_k)`: This tool performs case-based retrieval. The agent can issue a query, which could be derived from the given bug report or current problem-solving state, to find semantically related historical commits. We use BM25 for matching the query against the commit messages, as it is highly effective for the semi-structured, keyword-rich nature of commit messages. The interface returns a ranked list of the top-k relevant commits, including their unique IDs (commit SHAs), messages, and modified files in the commit patch.
- `ExamineCommit(id)`: Once a potentially relevant commit is identified, this tool allows the agent to "zoom in" and retrieve its full context based on its ID, including the complete code patch (in diff format) and any linked issues, providing a comprehensive view of the original problem and its corresponding solution.

By using these tools, the agent can ground its reasoning in historical precedent, leveraging past solutions as powerful exemplars to aid in its understanding of the codebase/problem and guide its investigation.

## 3.2 Semantic Memory of Active Code Functionality

**Memory Construction.** While episodic memory provides specific examples, semantic memory offers a generalized, high-level understanding of the codebase. The rationale is that files frequently modified in the recent past are "development hotspots"—areas that are either central to the repository's functionality or are undergoing active change. Research in defect prediction has long established that high code churn and frequent modifications correlate strongly with defect proneness (Nagappan & Ball, 2005; Zhao et al., 2023), making these files prime candidates for the agent's attention. We first analyze the commit history to identify the top-k most edited files, where k is much smaller than the total amount of files in the codebase. Then, for each of these files, we use an LLM to read its source code and distill its functionalities into a high-level natural language summary (details in Appendix A). This process creates a compact semantic knowledge base that maps critical files to their core responsibilities, focusing exclusively on the most dynamic parts of the repository.

**Memory Interfaces.** The agent accesses this knowledge base again through a simple query interface:

- `ViewSummary(file_name)`: This retrieves the cached summary for a specific file (if it exists in the memory), allowing the agent to quickly understand a file's purpose without needing to read its entire source code.
- `SearchSummary(query, top_k)`: This allows the agent to perform a keyword-based search over the entire collection of file summaries. It returns the top-k most relevant (file, summary) pairs, helping the agent to locate modules that are related to the issue or current exploration intent.

The semantic memory provides the agent with crucial architectural context, biasing its search towards more promising areas and preventing it from getting lost in the vast, irrelevant or stable parts of the codebase.

## 3.3 Integration with LocAgent

The memory tools are designed to be modular and can be straightforwardly integrated into existing agentic frameworks. In this work, we integrate them into **LocAgent**, a state-of-the-art localization framework that operates based on the ReAct paradigm (Yao et al., 2023). A LocAgent-powered agent

iteratively cycles through a "Thought, Act, Observation" loop. In the "Act" step, it synthesizes an API call to one of its available tools, whose execution feedback is returned to the agents via the next "Observation" entry. For context, LocAgent's core tools allow it to navigate a heterogeneous graph representation of the codebase:

- `SearchEntity`: Searches the codebase for entities matching a keyword query, typically serving as an entry point for exploration.
- `TraverseGraph`: Performs a multi-hop, type-aware breadth-first search from a starting entity to explore code relationships, which include 1) basic *contain* relationships between folders and files, 2) *invoke* relationships between functions and classes, 3) *import* relationships from files to functions/classes, and 4) the *inherit* relationship between classes.
- `RetrieveEntity`: Fetches the full source code and detailed information for a specific code entity (e.g., a file, class, or function).

Our integration simply expands the action space with the memory-based tools, as illustrated in Figure 1. Intuitively, the memory-based tools could nicely complement the existing toolset in LocAgent. For example, an agent can now use memory search tools to fetch related commits or files, combined with concrete examination of individual entries when necessary, to form an experience-based hypothesis. It can then use LocAgent's tools to perform a more detailed investigation of the code entities surrounding these candidates. This creates a powerful synergy between high-level, memory-guided direction and low-level, structural code analysis.

## 4 EXPERIMENTS

### 4.1 SETUP

**Datasets.** We evaluate our approach on two benchmarks: **SWE-bench-verified** (Jimenez et al., 2024), which contains 500 examples from 12 repositories, and the more recent **SWE-bench-live** (Zhang et al., 2025) benchmark. For SWE-bench-live, we use a high-quality subset created from the intersection of its 'lite' and 'verified' splits, filtering for instances requiring five or fewer files to be modified. This results in 130 examples across 62 repositories.

**Baselines.** We compare our method, **RepoMem**, against several state-of-the-art methods in different types of approaches:

- **CodeRankEmbed** (Suresh et al., 2025), a leading retrieval-based method leveraging large-scale training with careful data filtering and hard negative mining.
- **Agentless** (Xia et al., 2025), a leading procedural method that prompts an LLM with repository structure.
- **LocAgent** (Chen et al., 2025b), a state-of-the-art agentic framework for localization as discussed earlier, which our RepoMem method is built directly upon. This also allows for a direct comparison of the impact of integrating repository memory.

**Evaluation Metrics.** We evaluate file-level localization performance via **Accuracy@k** (following prior work (Chen et al., 2025b)), defined as the percentage of examples where the set of top-k predicted files completely covers the ground-truth files.

**Implementation Details.** All experiments use GPT-4o (2024-05-13) as the backbone LLM. For memory construction, we consider the 7,000 commits prior to the given issue's base commit, and identify the top 200 most frequently edited files for constructing the semantic memory. While we employ a fixed window size as a straightforward heuristic, the rich literature in JIT-SDP suggests that more sophisticated filtering strategies—such as time-decay, developer experience filtering, or co-change history (Zhao et al., 2023)—could further optimize memory efficiency and relevance. We leave the exploration of such domain-informed strategies to future work. Additional discussions on memory construction are provided in Appendix A.

### 4.2 MAIN RESULTS

Table 1 presents the main experimental results. We include additional results in Appendix B. **RepoMem** consistently outperforms baselines on both benchmarks. On SWE-bench-verified, RepoMem achieves an *Acc@5* of 76.5%, a 4.9% absolute improvement over the strong LocAgent baseline. The

Table 1: Main results on code localization benchmarks. RepoMem significantly outperforms all other methods across both benchmarks, demonstrating the effectiveness of incorporating repository memory. Both episodic and semantic memory components contribute positively, with their combination yielding the best performance.

| Methods | SWE-bench-verified | | | SWE-bench-live | | |
|---|---|---|---|---|---|---|
| | *Acc@1* | *Acc@3* | *Acc@5* | *Acc@1* | *Acc@3* | *Acc@5* |
| CodeRankEmbed (Suresh et al., 2025) | 29.6 | 45.1 | 54.3 | 26.2 | 44.6 | 52.3 |
| Agentless (Xia et al., 2025) | 53.3 | 67.8 | 71.4 | 40.0 | 60.0 | 62.3 |
| LocAgent (Chen et al., 2025b) | 64.8 | 70.4 | 71.6 | 59.2 | 60.8 | 63.1 |
| RepoMem (episodic-only) | 67.8 | 72.4 | 74.3 | 60.0 | 61.5 | 64.6 |
| RepoMem (semantic-only) | 65.0 | 71.0 | 72.8 | 56.9 | 61.5 | 63.9 |
| RepoMem | **68.6** | **74.5** | **76.5** | **60.8** | **63.9** | **66.2** |

Table 2: Per-repository performance comparison (Acc@5) on SWE-bench-verified. Repositories are sorted by the average number of historical commits available, where "others" is the union of repos with less than 10K commits. Our method sees strong gains in repositories with rich commit histories but can be hindered in those with limited or irrelevant history.

| Repo | matplotlib | sympy | astropy | django | scikit-learn | sphinx | pytest | others |
|---|---|---|---|---|---|---|---|---|
| # Examples | 34 | 73 | 22 | 231 | 32 | 44 | 18 | 46 |
| # Avg. Commits | 43.9K | 39.8K | 31.2K | 29.2K | 25.1K | 17.2K | 12.4K | 4.5K |
| LocAgent *Acc@5* | 76.5 | 69.9 | 86.4 | 72.3 | 93.8 | 47.7 | 61.1 | 67.4 |
| RepoMem *Acc@5* | 82.4 (+5.9) | 72.6 (+2.7) | 86.4 (+0.0) | 79.7 (+7.4) | 96.9 (+3.1) | 59.1 (+11.4) | 77.8 (+16.7) | 54.3 (-13.1) |

gains are also consistent on the more diverse SWE-bench-live dataset. Ablating on the effect of each memory, using only episodic memory ('episodic-only') provides a significant boost over LocAgent, demonstrating the value of referencing past commit history. Similarly, using only semantic memory ('semantic-only') also improves performance by helping the agent focus on actively developed parts of the codebase. The best results are achieved when both memory components are combined, indicating that they provide complementary information: episodic memory offers concrete solutions to similar past problems, while semantic memory provides high-level architectural context for the agent to leverage.

Table 2 provides a breakdown of performance by repository on SWE-bench-verified, sorted by the average number of historical commits available. The results reveal a clear correlation: repositories with a rich commit history benefit the most from RepoMem. This strongly supports our hypothesis that commit history is a valuable source for memory building. Conversely, for the "others" group which consists of repositories with limited history, performance degrades. This is likely because the memory contains too little relevant information, and the agent's exploration of this sparse history can be more distracting than helpful.

### 4.3 ANALYSIS

We perform a series of analyses of RepoMem on SWE-bench-verified, to gain deeper insights into the effect of integrating repository memory.

**Shift in Agent Behavior.** The introduction of memory drastically alters the agent's problem-solving strategy. As shown in Figure 3, agents equipped with the memory significantly reduce their reliance on exhaustive exploration tools (`TraverseGraph`) and direct code inspection (`RetrieveEntity`). This reflects a strategic shift from brute-force navigation to a more targeted, hypothesis-driven investigation, where the agent integrates its accumulated repository knowledge to form hypotheses, and performs detailed exploration/verification leveraging the original LocAgent tools—a process that more closely mirrors an experienced human developer's workflow.

**Efficiency Analysis.** We find that integrating the memory introduces a strategic cost-effectiveness trade-off instead of a uniform overhead. First, as shown in the cross-comparison in Table 3, the additional expenditure is primarily allocated to solving difficult problems—the most significant cost increase occurs in examples where LocAgent fails. This indicates that overall, our method

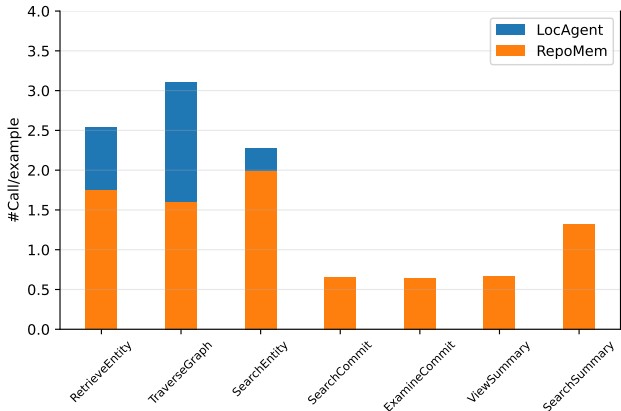

Figure 3: **Tool use distribution for LocAgent vs. RepoMem.** The introduction of memory-based tools drastically alters agent behavior. RepoMem significantly reduces its reliance on exhaustive exploration tools like `TraverseGraph` and direct code reading (`RetrieveEntity`), indicating a strategic shift from brute-force navigation to a more targeted, hypothesis-driven investigation guided by memory.

Table 3: **Cross-comparison of LLM API cost (LA: LocAgent, RM: RepoMem).** Each cell shows the average cost per example for LocAgent → RepoMem. The largest cost increase occurs in the bottom two quadrants, which are examples where LocAgent fails. This indicates that the additional cost is primarily a strategic investment to improve accuracy on difficult tasks.

|  | RM Succeeds | RM Fails |
|---|---|---|
| **LA Succeeds** | $0.58 → $0.68 | $0.59 → $0.66 |
| **LA Fails** | $0.54 → $0.89 | $0.59 → $0.87 |

Table 4: **Impact of retrieval method for memory interface on the performance of `django` repository.** Sparse retrieval using BM25 with a custom LLM-based tokenizer outperforms both a standard tokenizer and a strong dense retrieval model (GritLM-7B).

| Retrieval Methods | django/django | | |
|---|---|---|---|
| | Acc@1 | Acc@3 | Acc@5 |
| Dense retrieval | 65.8 | 71.9 | 73.6 |
| BM25 (whitespace) | 67.1 | 74.5 | 77.9 |
| BM25 (LLM) | 70.1 | 76.6 | 79.7 |

strategically invests additional resources to solve challenging problems that the baseline cannot, rather than spending wastefully on problems that could already be solved without resorting to the memory.

More interestingly, the cost impact is highly variable at the instance level. Figure 4 shows a scatter plot of per-example costs, again cross-comparing the two methods. While the average cost increases, the plot reveals high variance across the examples. For many problems, RepoMem is significantly cheaper than LocAgent (points far below the diagonal), likely because the memory provided a more direct hint to the solution. For some others, it could instead be much more expensive (points far above the diagonal), likely on problems where the memory proved fruitless and only added overhead and distractions. This heterogeneity highlights that average cost can be a misleading metric, and the efficiency of our memory-augmented agent is highly dependent on the relationship between the current problem and the repository's history.

These findings also suggest a promising future direction: training agents to be more strategic about when to use memory tools. An agent that could first assess a problem's novelty might learn to rely on memory for issues that are related to the past experience, while defaulting to first-principle explorations for unprecedented ones, optimizing the cost-effectiveness trade-off.

**Retrieval Methods for Memory Interfaces.** Here, we investigate the choice of retrieval method for our memory interfaces. We compare three approaches on the `django` repository, with results shown in Table 4. Here, we use the strong GritLM-7B model for dense retrieval (Muennighoff et al., 2025). Our default method in the main results uses BM25 with an LLM-based tokenizer that recognizes code entity names, which outperforms standard whitespace tokenization. More notably, sparse retrieval methods significantly outperform dense retrieval. We hypothesize this is due to the unique vocabulary

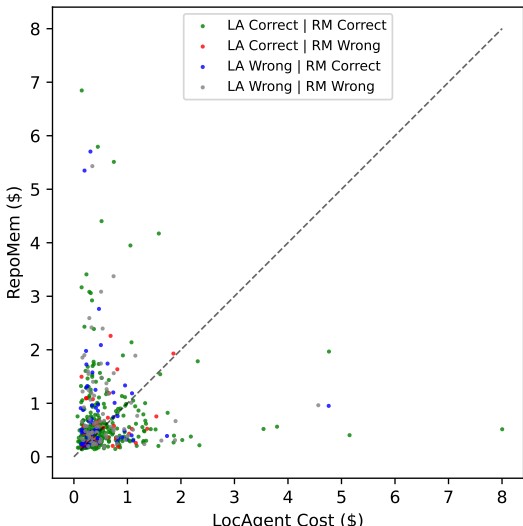

Figure 4: **Per-example cost comparison (LA: LocAgent, RM: RepoMem).** This scatter plot shows the LLM API cost for each example, where the $x$ and $y$ coordinates correspond to the cost of LocAgent and RepoMem, respectively. Points below the diagonal line indicate RepoMem was cheaper, while points above indicate it was more expensive. The high variance reveals that the efficiency impact of integrating memory is problem-dependent: it provides significant savings on some tasks but incurs overhead on others, a nuance missed by average cost metrics.

of code-related utterances in software repositories—for example, entities like 'MigrationWriter' and 'OperationWriter' may be semantically close but are functionally very distinct. Sparse retrieval methods, which rely on exact keyword matches, excel at handling this "rigid" vocabulary. Similar phenomena are also observed in prior work, e.g., Sciavolino et al. (2021) finds that dense retrievers could drastically underperform sparse methods in entity-centric question-answering.

**Error Analysis.** We conducted a small-scale analysis of the failure cases of RepoMem to better understand its limitations. As expected, the primary failure mode occurs when memory retrieval yields little useful information about the issue, a problem stemming from either the novelty of the issue or shortcomings in the retrieval methods. In such instances, the agent receives irrelevant information that can pollute its reasoning context and distract it—a well-known challenge for LLMs (Shi et al., 2023). This can lead to performance worse than the baseline, as observed in repositories with sparse histories (Table 2). These findings highlight promising directions for future work, such as designing/training better memory interfaces and developing mechanisms that enable the agent to dynamically decide whether to rely on the memory or instead fall back on first-principles reasoning (as discussed earlier).

## 5   CONCLUSION

In this work, we take an initial step toward addressing a key limitation of current language agents for software engineering: their lack of long-term repository memory. We propose a simple yet effective solution that leverages the rich contextual information embedded in a repository's commit history. By building two complementary memory stores—an episodic memory of past commits and linked issues, and a semantic memory of active code functionality—we enable agents to draw on past knowledge when tackling future tasks. Our experiments show that this memory-augmented approach substantially improves code localization performance on established benchmarks. Further analysis reveals a shift in agent behavior toward a more experience-guided strategy that better reflects human expertise. Overall, this work underscores the importance of incorporating long-term memory into agent workflows, paving the way for more capable and experienced software engineering assistants.

ETHICS STATEMENT

All authors of this paper have read and adhered to the ICLR Code of Ethics. Our research is built upon publicly available datasets, which are derived entirely from open-source software repositories. The study does not involve human subjects, and our data processing steps do not introduce any new ethical concerns regarding privacy, bias, or fairness. The proposed methods are designed for software engineering assistance and do not present foreseeable risks of misuse or negative societal impact.

REPRODUCIBILITY STATEMENT

We have made every effort to ensure our work is reproducible. Our experiments are conducted on the public SWE-bench-verified and SWE-bench-live benchmarks. The methodology for constructing the episodic and semantic memory components is detailed in §3, and the implementation details are provided in §4.1. To further facilitate replication, we provide comprehensive documentations, examples, and prompts used for our agent in Appendix C and Appendix D. The source code for our framework and experiments will also be made publicly available upon publication.

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

## A  MEMORY CONSTRUCTION AND OPERATIONAL COSTS

### A.1  FURTHER DETAILS

To construct the semantic memory, for each of the most frequently edited files, we employ the LocAgent framework tasked with summarizing the file's functionality, where the model could actively explore dependencies across different files whenever it determines that local context is insufficient. This helps make the generated summaries context-aware and accurate. On average, each file costs \$0.23 (calculated via GPT-4o input/output pricing).

Our episodic memory construction currently relies on explicit links between commits and issues (e.g., commit messages containing *"Fixes #123"*) to retrieve the full context of past solutions. We acknowledge that such explicit links can be noisy or missing, a limitation well-documented in existing literature (Bachmann et al., 2010). The absence of these links potentially lowers the recall of our memory retrieval. Future work could explore mitigating this limitation by adopting more robust linking strategies used in JiT-SDP research (McIntosh & Kamei, 2018; Trautsch et al., 2020; Quach et al., 2021; Zhao et al., 2023).

### A.2  OVERHEAD

**Offline Construction.** We measured the wall-clock time on `SymPy`, the repository with the largest history in SWE-bench-Verified (containing over 60k historical commits). The complete memory construction took approximately 45 minutes: roughly 37 minutes for data crawling (commits, issues, etc.) and 8 minutes for parallelized preprocessing and semantic memory generation. Importantly, this is a one-time setup cost per repository.

**Online Latency.** To assess the runtime impact during agent execution, we analyzed a random 50-example subset of SWE-bench-Verified. The average total runtime per instance was 28.6 seconds. Crucially, the overhead introduced by our memory search tools is negligible, averaging only 0.22 seconds per instance (relatedly, the standard LocAgent tools cost 0.37 seconds). The vast majority of the runtime is dominated by LLM inference, confirming that incorporating repository memory adds minimal latency to the agentic loop. We also collect the average response length (in tokens) for memory tools compared to standard LocAgent tools across SWE-bench-Verified instances. The memory tools are generally at similar levels of token lengths compared with LocAgent tools:

- LocAgent Tools: RetrieveEntity (7.10K), TraverseGraph (2.98K), SearchEntity (1.66K).
- Memory Tools: SearchCommit (2.11K), ExamineCommit (1.81K), ViewSummary (1.67K), SearchSummary (6.18K).

## B  ADDITIONAL RESULTS

### B.1  SENSITIVITY TO $K$ IN SEMANTIC MEMORY CONSTRUCTION

We investigate the impact of the hyperparameter $K$ (the number of top active files included in Semantic Memory) on the localization effectiveness. The experiments are done on the Django repository, which is our primary repository for preliminary analysis and prototyping.

**Coverage.**  While serving mostly as a proxy, we analyzed the coverage of target (buggy) files within the top-$K$ frequently edited files. As shown in Table 5, active files demonstrate high overall coverage of target files.

Table 5: Coverage of target (buggy) files within the top-$K$ frequently edited files in the Django repository.

| $K$ **(Top Active Files)** | 50 | 100 | 200 | 300 | 400 |
|---|---|---|---|---|---|
| **Coverage (%)** | 39.4 | 62.3 | 83.1 | 87.4 | 90.9 |

**Performance Sensitivity.**  We evaluated the localization performance (Acc@5) across different values of $K$. As presented in Table 6, performance improves monotonically with $K$, but exhibits diminishing returns as $K$ increases.

Table 6: Impact of $K$ on localization performance (Acc@5) on the Django repository.

| $K$ **(Top Active Files)** | 50 | 100 | 200 | 300 | 400 |
|---|---|---|---|---|---|
| **Acc@5 (%)** | 74.4 | 76.2 | 79.7 | 80.5 | 81.0 |

Based on these results, we selected $K = 200$ as the optimal setting to balance retrieval performance and computational cost. We maintain this value across all repositories for simplicity, though $K$ could also be treated as a repo-specific hyperparameter tuned to the size or activity level of each repository.

### B.2  IMPACT ON ISSUE RESOLUTION RATE

To assess the downstream impact of our localization improvements, we conducted an end-to-end evaluation on SWE-bench-verified. We utilized the patch generation pipeline from Agentless (Xia et al., 2025) under standard settings with GPT-4o as the backbone. Specifically, the pipeline employs LLMs to generate 10 candidate patches per issue based on localization results, re-ranks them using regression and reproduction tests, and selects the top patch for evaluation.

As shown in Table 7, improved localization accuracy translates directly to higher resolution rates, with RepoMem (Full) outperforming baseline methods. This also confirms that localization performance strongly correlates with the final resolve rate, a trend consistent with prior findings (Xia et al., 2025; Chen et al., 2025b).

Table 7: Localization accuracy and issue resolution rate on SWE-bench-verified.

| Method | Loc Acc@5 | Resolve Rate |
|---|---|---|
| CodeRankEmbed (Suresh et al., 2025) | 54.3 | 26.8 |
| Agentless (Xia et al., 2025) | 71.4 | 36.2 |
| LocAgent (Chen et al., 2025b) | 71.6 | 37.0 |
| RepoMem (Episodic-only) | 74.3 | 38.8 |
| RepoMem (Semantic-only) | 72.8 | 37.6 |
| **RepoMem (Full)** | **76.5** | **40.4** |

### B.3  RETRIEVAL ABLATION

Here, we include the full comparison between the LLM-based tokenizer and the standard BM25 tokenizer utilized in the memory retrieval interfaces across different repositories. The results (Table 8) confirm that the LLM-based tokenizer consistently outperforms the standard tokenizer across the majority of repositories.

Table 8: Localization performance (Acc@5) under Standard BM25 and LLM-based tokenizer in memory retrieval.

| Method | Matplotlib | Sympy | Astropy | Django | Sklearn | Sphinx | Pytest | Others |
|---|---|---|---|---|---|---|---|---|
| BM25 (Standard) | 79.4 | 69.9 | 81.8 | 77.9 | 90.6 | 59.1 | 72.2 | 47.8 |
| BM25 (LLM-based) | 82.4 | 72.6 | 86.4 | 79.7 | 96.9 | 59.1 | 77.8 | 54.3 |

## C  DOCUMENTATION AND EXAMPLE RESPONSES OF MEMORY TOOLS

Figure 5, 6, 7, 8 show the documentation and example responses of the memory tools.

*SearchCommit*

**Doc:**

This tool searches the repository's commit history to find commits similar to a given query - typically a hypothetical commit message. It uses the BM25 ranking algorithm to match the input against historical commit messages, surfacing relevant past changes. Using this tool early in the investigation is recommended, as it can help identify related issues or changes from the past, allowing you to quickly locate potential areas in the codebase to focus on.

**Parameters**: {
    'type': 'object',
    'properties': {
        'query_list': {
            'type': 'array',
            'items': {'type': 'string'},
            'description': (
                "A list of queries. Each query can be a hypothetical commit message, which would be matched with the history commit
messages to retrieve similar commits."
            )
        },
        'top_k': {
            'type': 'integer',
            'description': (
                "top_k commits will be returned."
            ),
            'default': 20
        }
    },
    'required': ['query_list'],
}

**Example API call:** *SearchCommit(query='Fix Cannot drop unique_together constraint on a single field with its own unique=True constraint', top_k=20)*

**Observation:**

##Searching for query `Fix Cannot drop unique_together constraint on a single field with its own unique=True constraint`...
### Search Result:
SHA: 73a6ab638 | Commit message: Fixed #25551 -- Fixed migration operations ordering when adding fields and a
unique_together constraint. | Edited files: ['django/db/migrations/autodetector.py']
SHA: 533b20877 | Commit message: Fixed #29224 -- Fixed removing index_together indexes if exists unique_together constraint
on the same fields. | Edited files: ['django/db/backends/base/schema.py']
SHA: f5a324391 | Commit message: Fixed #33125 -- Avoided redundant unique constraint when converting a non-unique field to
primary key on MySQL and PostgreSQL. | Edited files: ['django/db/backends/base/schema.py',
'django/db/backends/oracle/schema.py']
...
================================================================================================
*ExamineCommit*

**Doc:**

This tool examines the details of a commit based on the short (9-digit) commit sha, including the commit patch and optionally the issue description.
Note that the line numbers displayed in the patch are from those history commits, not the current repo you are working on.

**Parameters**: {
    'type': 'object',

Figure 5: Documentation and example outputs from the memory tools.

```
'properties': {
    'sha_list': {
        'type': 'array',
        'items': {'type': 'string'},
        'description': (
            "A list of short 9-digit commit sha that you wish to examine."
        )
    },
    'display_issue': {
        'type': 'boolean',
        'description': (
            "Set to True if you wish to include the issue description in the output."        ),
        'default': False
    }
},
'required': ['sha_list'],
}
```

**Example API call:** *ExamineCommit(sha=['50931dfa5'], display_issue=True)*
**Observation:**
##Commit for sha `50931dfa5`...
issue summary: Allow management commands to check if database migrations are applied
issue description: When creating a new project, you can sometimes forget to run `manage.py migrate` before creating the initial superuser (especially if you don't execute `runserver` before, which display a warning about migrations not applied). The resulting error make sense, it can't access to auth_user, since it does not exist yet:
    {{{
    $ django-admin.py startproject sample
    $ cd sample/ && python manage.py createsuperuser
    ...
    ... but with a little try/except, it could be nicer and give a more meaningful information:

    {{{
    $ python manage.py createsuperuser
    CommandError: You must execute `manage.py migrate` once before creating a super user
    }}}
    I have a patch ready (as I said, it just a try/except, see attachment) but, if accepted, it would need unit tests to
be complete.
Edited file: django/contrib/auth/management/commands/changepassword.py
Patch:
```
@@ -12,7 +12,7 @@
 class Command(BaseCommand):
    help = "Change a user's password for django.contrib.auth."
-
+    requires_migrations_checks = True
    requires_system_checks = False
    def _get_pass(self, prompt="Password: "):
----
```
Edited file: django/contrib/auth/management/commands/createsuperuser.py
Patch:
```
@@ -23,6 +23,7 @@ class NotRunningInTTYException(Exception):
 class Command(BaseCommand):
```

Figure 6: Documentation and example outputs from the memory tools.

```
    help = 'Used to create a superuser.'
+   requires_migrations_checks = True
    def __init__(self, *args, **kwargs):
        super(Command, self).__init__(*args, **kwargs)
```

==========================================================================================

***ViewSummary***

**Doc:**

This tool views summary of python files in the repo. You can input a list of file names you wish to examine.

**Parameters**: {
   'type': 'object',
   'properties': {
     'file_list': {
       'type': 'array',
       'items': {'type': 'string'},
       'description': (
         "A list of filenames that you wish to view the summaries."
       )
     },
   },
   'required': ['file_list'],
}

**Example API call:** *ViewSummary(file_list=["django/db/migrations/serializer.py"]):*

**Observation:**

##Summary for file `django/db/migrations/serializer.py`...

The `serializer.py` module in Django's migration system provides various serializer classes to convert Python objects into a serialized form suitable for migration files. The module includes a base class `BaseSerializer` and multiple subclasses, each handling specific types of objects such as lists, dictionaries, datetime objects, models, and custom functions. Each serializer implements a `serialize()` method that returns a string representation of the object and necessary import statements. The `Serializer` class maintains a registry of type-to-serializer mappings, allowing easy registration and usage of serializers through the `serializer_factory` function. This ensures objects are properly serialized when creating or applying migrations in Django.

==========================================================================================

***SearchSummary***

**Doc:**

This tool searches the repository's cached file summaries to find files similar to a given query - typically a message describing some functionalities or keywords. It uses BM25 to match the input query against file summaries and returns top-k ones.

**Parameters**: {
   'type': 'object',
   'properties': {
     'query': {
       'type': 'string',
       'description': (
         "A query, which can be a message describing some functionalities or keywords, to be matched with the available file
summaries to retrieve similar ones."
       )
     },
     'top_k': {

Figure 7: Documentation and example outputs from the memory tools.

```
            'type': 'integer',
            'description': (
                "the top_k files to return."
            ),
            'default': 5
        }
    },
    'required': ['query'],
}
```

**Example API call:** *SearchSummary(query='*generates a migration file that is missing an import statement*', top_k=4):*
**Observation:**
## Searching for query `generates a migration file that is missing an import statement`...
### Search Result:
file: django/db/migrations/operations/special.py | summary: The `special.py` file in `django/db/migrations/operations` defines three classes for managing database migrations in Django:\n\n1. **`SeparateDatabaseAndState`**: This class separates migration operations into those affecting the database and those affecting the state. It allows for operations that do not support state change to still apply the changes or vice versa. It includes methods to deconstruct operations, forward and backward migrations specifically for state and database, and a description of the combined state/database change.\n\n2. **`RunSQL`**: This class is used to run raw SQL statements during migrations. It supports providing a reverse SQL statement for reversible migrations and can handle state changes represented by state operations. It includes methods for deconstructing the operations, performing forward and backward migrations with SQL, and a description of the raw SQL operation.\n\n3. **`RunPython`**: This class facilitates running custom Python code during migrations, suitable for versioned ORM operations. It accepts a callable for forward migrations and an optional callable for reverse migrations. It includes methods to deconstruct the operation, perform forward and backward database migrations, and a description of the raw Python operation. It also ensures all models are reloaded to accommodate possible delays.\n\nEach class inherits from `Operation`, ensuring compatibility with Django's migration framework and providing essential interfaces for migration processes.

file: django/db/migrations/serializer.py | summary: The `serializer.py` module in Django's migration system provides various serializer classes to convert Python objects into a serialized form suitable for migration files. The module includes a base class `BaseSerializer` and multiple subclasses, each handling specific types of objects such as lists, dictionaries, datetime objects, models, and custom functions. Each serializer implements a `serialize()` method that returns a string representation of the object and necessary import statements. The `Serializer` class maintains a registry of type-to-serializer mappings, allowing easy registration and usage of serializers through the `serializer_factory` function. This ensures objects are properly serialized when creating or applying migrations in Django.

file: django/core/management/commands/makemigrations.py | summary: This Django management command, `makemigrations.py`, is used to create new database migration files for specified apps. It offers various options such as performing a dry run, merging migration conflicts, creating empty migrations, and controlling verbosity. It ensures consistent migration history across databases, checks for migration conflicts, and handles user prompts interactively or non-interactively. The script generates migration files based on detected model changes, writes them to disk, and can display the details for review. It also includes functionality for merging conflicting migrations interactively, ensuring consistency and resolving dependencies.

file: django/core/management/sql.py | summary: ...

Figure 8: Documentation and example outputs from the memory tools.

## D  AGENT PROMPT

We use the same task instruction prompt from the original LocAgent framework to guide the agent, which is displayed in Figure 9 for completeness.

```
Prompt

Given the following GitHub problem description, your objective is to localize the specific files, classes or functions, and lines
of code that need modification or contain key information to resolve the issue.

Follow these steps to localize the issue:
## Step 1: Categorize and Extract Key Problem Information
- Classify the problem statement into the following categories:
  Problem description, error trace, code to reproduce the bug, and additional context.
- Identify modules in the '{{package_name}}' package mentioned in each category.
- Use extracted keywords and line numbers to search for relevant code references for additional context.

## Step 2: Locate Referenced Modules
- Accurately determine specific modules
  - Explore the repo to familiarize yourself with its structure.
  - Analyze the described execution flow to identify specific modules or components being referenced.
- Pay special attention to distinguishing between modules with similar names using context and described execution flow.
- Output Format for collected relevant modules:
  - Use the format: 'file_path:QualifiedName'
  - E.g., for a function `calculate_sum` in the `MathUtils` class located in `src/helpers/math_helpers.py`, represent it as:
    'src/helpers/math_helpers.py:MathUtils.calculate_sum'.

## Step 3: Analyze and Reproducing the Problem
- Clarify the Purpose of the Issue
  - If expanding capabilities: Identify where and how to incorporate new behavior, fields, or modules.
  - If addressing unexpected behavior: Focus on localizing modules containing potential bugs.
- Reconstruct the execution flow
  - Identify main entry points triggering the issue.
  - Trace function calls, class interactions, and sequences of events.
  - Identify potential breakpoints causing the issue.
  Important: Keep the reconstructed flow focused on the problem, avoiding irrelevant details.

## Step 4: Locate Areas for Modification
- Locate specific files, functions, or lines of code requiring changes or containing critical information for resolving the issue.
- Consider upstream and downstream dependencies that may affect or be affected by the issue.
- If applicable, identify where to introduce new fields, functions, or variables.
- Think Thoroughly: List multiple potential solutions and consider edge cases that could impact the resolution.

## Output Format for Final Results:
Your final output should list the locations requiring modification, wrapped with triple backticks ```
Each location should include the file path, class name (if applicable), function name, or line numbers, ordered by importance.
Your answer would better include about 5 files.

### Examples:
```
full_path1/file1.py
line: 10
class: MyClass1
function: my_function1

full_path2/file2.py
line: 76
function: MyClass2.my_function2

full_path3/file3.py
line: 24
line: 156
function: my_function3
```

Return just the location(s)
Note: Your thinking should be thorough and so it's fine if it's very long.
```

Figure 9: The task instruction prompt used to guide the agent's reasoning process.

## E  USE OF LARGE LANGUAGE MODELS

Large Language Models (LLMs) were used solely as a general-purpose writing aid in the preparation of this paper. Specifically, they were used to help polish grammar and improve the clarity of certain sentences. No LLMs were used for research ideation, experimental design, data analysis, or drawing conclusions. All substantive contributions to the research and writing were made by the authors.

