# OpenReview forum: "Improving Code Localization with Repository Memory"
_ICLR.cc/2026/Conference — ICLR 2026 Poster_

### Official Review · Reviewer_bN9m · 2025-10-21

**Soundness:** 2
**Presentation:** 3
**Contribution:** 2
**Rating:** 4
**Confidence:** 3

**Summary:**

The paper proposes RepoMem, a lightweight repository memory for code localization agents, built from each repo’s commit history. It instantiates two complementary, non-parametric memories: (1) Episodic—a searchable store of prior commits and linked issues, queried via SearchCommit / inspected via ExamineCommit; and (2) Semantic—LLM-generated functionality summaries for frequently edited (“hotspot”) files, accessed via ViewSummary / SearchSummary. Integrated into LocAgent, RepoMem improves localization on SWE-bench-verified and a filtered SWE-bench-live subset, using GPT-4o (2024-05-13) as the backbone and reporting Accuracy@k.

**Strengths:**

The paper introduces a practical repository-memory mechanism for SWE-bench localization, combining (i) an episodic store built from pre-issue commit history and linked artifacts that can be searched/inspected at inference time, and (ii) a semantic memory of LLM-generated summaries for frequently edited (“hotspot”) files; the design is simple to integrate, and aligns with how human developers leverage history to narrow the search scope. In extensive evaluations, the memory delivers consistent performance gains over a strong LocAgent baseline, improving Accuracy@k on SWE-bench-verified and a live subset, with ablations indicating complementary benefits of episodic + semantic memories.

**Weaknesses:**

1. Limited Novelty in Core Mechanism: The agent's memory system, while presented as a compositional structure, relies heavily on established prompting techniques and existing memory concepts rather than introducing fundamentally new mechanisms for agent memory or reasoning. The contribution seems more focused on engineering specific prompts for memory management than on novel agent capabilities.
2. Ablation gaps (end-to-end impact). Empirics focus on localization Acc@k, but the paper does not quantify downstream SWE-bench end-to-end effects (e.g., Resolved@1, test pass rate, patch compile/apply rate, tool-call path length, wall-clock). Without E2E deltas versus a strong baseline (and versus LocAgent without memory), it is hard to assess whether memory truly improves fix outcomes.

**Questions:**

1. End-to-end impact: Can you report SWE-bench end-to-end results (resolution rate, test pass) with/without RepoMem, and per-component ablations, to quantify how localization gains translate to actual fixes?
2. Cost/latency & token accounting: What are (a) average prompt tokens per component, (b) offline token cost to build semantic memory (top-200 file summaries), and (c) online latency/$ overhead relative to LocAgent?

---

> ### Author Response · Authors · 2025-11-21
>
> We thank the reviewer for the thoughtful comments and for recognizing our repository memory design as "practical," "simple to integrate," and aligned with human developer workflows. We are encouraged that you found our evaluation extensive. We address your concerns and questions below.
>
> **1. Novelty in Core Mechanism**
>
> We acknowledge the reviewer's observation that our work leverages existing prompting techniques rather than introducing new model architectures/designs. However, we believe the novelty and contribution of this work lie in identifying and addressing a critical missing component in current agentic workflows for coding: the lack of long-term repository memory.
> - **Bridging the Gap**: Current state-of-the-art agents (e.g., LocAgent, Agentless) are effectively "memoryless", treating every issue as a fresh puzzle. In contrast, experienced developers strongly rely on long-term memory accumulated from the repository’s evolution. We take a first step to bridge this gap.
> - **Conceptual Structure**: While the implementation utilizes standard agentic frameworks, the conceptual contribution is the novel structuring of commit history and functionality summaries into distinct non-parametric memory stores (Episodic and Semantic). This serves as a first concrete step toward coding agents that accumulate experience over time.
> - **Effectiveness via Simplicity**: We view the lightweight, compositional nature of RepoMem as a significant strength. It demonstrates that complex architectures are not strictly necessary to achieve substantial gains; rather, identifying the appropriate memory abstractions (episodic vs. semantic) is the key driver of performance
>
> **2. End-to-End Impact**
>
> To address the concern regarding downstream impact, we conducted an end-to-end evaluation on SWE-bench-verified. We utilized the patch generation pipeline from Agentless (Xia et al., 2025) under standard setups (with GPT-4o as backbone). Concretely, Agentless employs LLMs to generate 10 candidate patches per issue given the localization results, then re-rank the patches using both regression and reproduction tests, and select the top patch to evaluate the final resolve rate.
>
> **Results**: As shown in the table below, better localization directly translates to higher resolution rates, where RepoMem outperforms other baselines. As can be seen, the localization performance strongly correlates with the final resolve rate, which is also consistent with prior findings (Chen et al., Xia et al.).
>
> | Method | Localization Acc@5 | Resolve Rate |
> | :--- | :--- | :--- |
> | CodeRankEmbed (Suresh et al., 2025) | 54.3 | 26.8 |
> | Agentless (Xia et al., 2025) | 71.4 | 36.2 |
> | LocAgent (Chen et al., 2025b) | 71.6 | 37.0 |
> | RepoMem (episodic-only) | 74.3 | 38.8 |
> | RepoMem (semantic-only) | 72.8 | 37.6 |
> | RepoMem (Full) | **76.5** | **40.4** |
>
> **3. Cost and Latency**
>
> **(a) Average prompt tokens per component**: below is the average response length (in tokens) for memory tools compared to standard LocAgent tools across SWE-bench-verified instances. The memory tools are generally at similar levels of token lengths compared with LocAgent tools.
> - LocAgent Tools: RetrieveEntity (7.10K), TraverseGraph (2.98K), SearchEntity (1.66K).
> - Memory Tools: SearchCommit (2.11K), ExamineCommit (1.81K), ViewSummary (1.67K), SearchSummary (6.18K).
>
> **(b) Offline token cost**: the cost to build the semantic memory is $0.23 per file (calculated using GPT-4o input/output pricing). Note that this is a one-time cost.
>
> **(c) Online latency/$ overhead**: we performed a fine-grained analysis of this in the paper (Lines 416-433 and Figure 4). The overhead is not uniform and exhibits high variance across examples: for some "easy" instances, RepoMem actually significantly *reduces* cost/latency by providing historical shortcuts to the solution. For "hard" instances (where LocAgent typically fails), RepoMem invests more tokens to explore the memory. Consequently, average cost metrics can be somewhat misleading - the additional cost represents a strategic investment in solving difficult problems that baseline agents cannot solve, rather than wasteful overhead.
>
> We thank the reviewer again for the thoughtful comments and constructive suggestions. We will incorporate the improvements and additional results into the revised draft over the next few days. We look forward to addressing any additional questions or comments.
>
> **References**
> - Chen et al. LocAgent: Graph-Guided LLM Agents for Code Localization. ACL-25
> - Xia et al. Agentless: Demystifying LLM-based Software Engineering Agents. Proceedings of the ACM on Software Engineering, 2025.

---

> > ### Author Response · Authors · 2025-11-24
> > **Help check the rebuttal**
> >
> > Dear Reviewer bN9m, we would greatly appreciate it if you could review our rebuttal and share your feedback. Please let us know if further clarification or additional responses are needed.

---

### Official Review · Reviewer_tvHt · 2025-11-01

**Soundness:** 2
**Presentation:** 3
**Contribution:** 2
**Rating:** 4
**Confidence:** 3

**Summary:**

The paper aims to enhance the fault localization agent with two types of memory designs.

The intuition is clear and well justified: A human developer would not consider a code repo as a fresh problem instance. Instead, she will consider the commit histories and other background knowledge about the code repo.

Specificially, this work introduces two types of memories:
1. A memory that summarizes key information in commit histories
2. A memory that summarizes the code semantics of some key files.

To identify the key files, the proposed system identifies the files that are most frequently modified in previous commits.
then they use LLM to generate summaries for those key files.

To enhance the localization agent system, the proposed memories are provided as tools to the agent.

The paper evaluates the localization agent on two issue fixing benchmarks: swe-bench and swe-bench-live. With the augmented tools, the agent demonstrates better performance in fault localization on those two benchmarks.

**Strengths:**

The paper targets an important problem and proposes a neat and intuitive solution.

**Weaknesses:**

Q1: Justify the active code identification. The proposed approach identifies frequently modified code files and only summarizes the semantics of those files. I think it is a strong prior to assume most issues only happens in those files. Are there any statistical support for this observation/heuristics? Are they specific to those two benchmarks?

Q2: Clarify the details about code summary generation. How the code summaries are generated? Many code snippets/functions have complex dependences across files. Do the generated summary consider the contexts across different files?

Q3: The proposed technique summarizes the top_k active edited file. Is the system performance sensitive to the selection of k?

Q4: Adaptation to newer commits. When the code repo receives new commits, how the pre-generated memories would be updated? For example, some of the code summaries may be out-dated.

**Questions:**

Please see above.

---

> ### Author Response · Authors · 2025-11-21
>
> We thank the reviewer for the thoughtful review and for recognizing our intuition regarding repository memory as "clear and well justified." We appreciate the constructive questions regarding our design choices and experimental analysis. Below, we address the specific concerns/questions raised.
>
> **Q1: Justification of active code identification**
>
> It's actually not necessary for the error source to be within these “active” files for the memory to be effective. Intuitively, frequently edited files represent natural developmental "hotspots" of the repository (i.e., areas repeatedly exposed to human maintainers). These files should provide strong cues that aid the LLM agent in locating the error source, even if they do not contain the bug themselves. On the other hand, we do have some statistics on the Django repo (our primary repository for preliminary analysis and prototyping) about the coverage of target (buggy) files across instances within the top-K frequently edited files:
>
> | K (Top Active Files) | 50 | 100 | 200 | 300 | 400 |
> | :--- | :--- | :--- | :--- | :--- | :--- |
> | **% of covered instances** | 39.4% | 62.3% | 83.1% | 87.4% | 90.9% |
>
> It could be seen that these active files indeed have a high overall coverage. Still, the coverage here is mostly a “proxy”; we believe the best quantitative support for our design choice is the final performance gain observed across different benchmarks/repos when augmenting the agent with semantic memory built from these active files. Regarding generalization to other repos, while we believe this intuition holds broadly, quantitatively proving this is challenging due to the lack of alternative high-quality benchmarks.
>
> **Q2: Clarification of code summary generation**
>
> Thank you for pointing out this missing detail. To generate file summaries, we directly employ the LocAgent framework tasked with summarizing the file's functionality, where the model could actively explore dependencies across different files whenever it determines that local context is insufficient. This helps make the generated summaries context-aware and accurate. We will clarify this in the revised draft.
>
> **Q3: Is the system performance sensitive to the selection of $K$?**
>
> We experimented with different values of $K$ on the Django subset. The results indicate that performance improves monotonically with $K$, but with diminishing returns as $K$ becomes large:
>
> | K | 50 | 100 | 200 | 300 | 400 |
> | :--- | :--- | :--- | :--- | :--- | :--- |
> | **Acc@5 (%)** | 74.4 | 76.2 | **79.7** | 80.5 | 81.0 |
>
> We selected $K=200$ as a sweet spot that balances performance and cost (we also used the same $K$ across repos for simplicity). With more sufficient resources, $K$ could also be treated as a repo-specific hyperparameter tuned to the size or activity level of each repository.
>
> **Q4: Adaptation to newer commits and updating pre-generated memories**
>
> This is a very insightful point regarding the "continual learning" aspect of the system. While current benchmarks operate on static snapshots, preventing us from experimentally simulating live evolution, our system could be naturally extended to handle such repo updates:
> - **Episodic Memory**: New commits and related issues could be simply appended to the searchable database.
> - **Semantic Memory**: We do not need to regenerate summaries from scratch. Instead, we can employ the LLM agent to "patch" the existing summary by providing it with the old summary and the “diff” of the new codebase changes. This incremental update could make the process computationally efficient.
>
> We thank the reviewer again for the thoughtful comments and constructive suggestions. We will incorporate the improvements and additional results into the revised draft over the next few days. We look forward to addressing any additional questions or comments.

---

> > ### Comment · Reviewer_tvHt · 2025-11-24
> >
> > Thank the authors for the discussion. The response addresses most of my concerns. I increased my rating to 6. Please include the discussions to the paper/appendix.

---

> > > ### Author Response · Authors · 2025-11-24
> > > **Thanks for the recognition**
> > >
> > > Thanks for your recognition. We will include the discussion in the final camera ready version.

---

### Official Review · Reviewer_Fbe2 · 2025-11-01

**Soundness:** 3
**Presentation:** 2
**Contribution:** 3
**Rating:** 6
**Confidence:** 4

**Summary:**

The paper augments repository-level localization agents with repository memory built from pre-issue commit histories—an episodic memory of past commits/issues and a semantic memory of summaries for frequently edited files—yielding consistent gains over LocAgent on SWE-bench-Verified.

**Strengths:**

- The framing—that human-like “long-term repository memory” is missing in current agents—is clear and well motivated, and the two concrete tools are simple yet effective: SearchCommit / ExamineCommit (BM25 over commit messages + diffs) and SearchSummary / ViewSummary (LLM summaries of “hot” files). Integration into LocAgent’s ReAct loop is clean and modular, with sensible leakage controls (time-slicing to pre-issue commits and filtering text overlaps).

- On SWE-bench-Verified, RepoMem improves Acc@5 from 71.6 → 76.5 (+4.9 abs) over LocAgent; on SWE-bench-Live, 63.1 → 66.2. The setup details (GPT-4o, 7k prior commits, top-200 files for summaries) are reported, and both episodic- and semantic-only ablations are positive, suggesting complementary value.

- Insightful analyses. (i) Per-repo analysis shows larger gains where commit histories are rich; (ii) cost analysis reveals instance-level heterogeneity—memory sometimes reduces cost dramatically but can add overhead on hard cases; (iii) retrieval study finds BM25 (with an LLM-aware tokenizer) outperforms a strong dense retriever (GritLM-7B) on Django. These help practitioners reason about when/why the method helps.

**Weaknesses:**

- Sensitivity to history density. Benefits correlate with repository commit volume; performance degrades on the “others” bucket with sparse histories (−13.1 Acc@5 vs. LocAgent), underscoring brittleness when history is limited or off-target. Some mitigation is discussed qualitatively but not implemented.


- Dataset selection and external validity. For SWE-bench-Live, the paper evaluates an intersectioned subset (Lite and Verified) and filters to ≤5 modified files (130 examples/62 repos). This practical choice is understandable but may bias toward easier issues; a sensitivity analysis to the filter would strengthen claims.


- Reporting gaps on efficiency & scalability. While costs are compared, the paper lacks index/build-time, memory footprint, and latency for constructing/searching the 7k-commit memory and top-200 summaries—numbers crucial for monorepos and CI integration.

**Questions:**

Retrieval extensions: given BM25’s strong showing on Django, report whether the LLM-aware tokenizer generalizes across repos and whether hybrid sparse+dense retrieval helps when commit phrasing diverges from issue wording

---

> ### Author Response · Authors · 2025-11-21
>
> We appreciate your positive assessment of our work, particularly for highlighting the motivation behind "long-term repository memory" and the modular integration of our tools. We are encouraged that you found our cost and per-repo analyses insightful. We address your concerns and questions below.
>
> **1. Sensitivity to History Density (Weakness 1)** We agree and acknowledge in the draft (e.g., lines 343-345) that the performance dependence on history density is a limitation of our approach, and this is a characteristic inherent to all experience-based systems/methods. For "memory-sparse" repositories (e.g., those in the "others" category), the memory tools could return lots of irrelevant information, acting as distractors that lead to the observed performance degradation. The main issue is that we are currently prompting an off-the-shelf LLM, which has little idea about what the memory contains. As noted in the paper, this suggests a future direction of training agents to use memory tools more strategically, and fall back to first-principle reasoning when the issue at hand is novel/the memory is sparse.
>
> **2. SWE-bench-Live Dataset Selection (Weakness 2)** We appreciate the comment on the filtering criteria. Our primary evaluation remains the full SWE-bench-Verified dataset, which is a well-established, high-quality benchmark. We included SWE-bench-Live to evaluate generalization on newer issues. Our rationale for selecting this specific subset within SWE-bench-Live is twofold:
> - **Quality Control**: SWE-bench-Live is a very recent benchmark. Such automatically constructed benchmarks are often prone to containing low-quality samples or noise, which is why we focused on the higher-quality subset found at the intersection of "Verified" and "Lite".
> - **Distribution Alignment**: The filtering for $\le 5$ files was chosen to align the evaluation distribution with the hyperparameter/prompt configuration of our backbone framework (LocAgent) and other baselines. These baselines are all optimized on SWE-bench-Lite or Verified, where nearly all issues involve $\le 5$ target files. Testing on issues with massive file changes would introduce a distribution shift for the backbone/baseline frameworks, making it difficult to isolate the specific contribution of our approach.
>
> **3. Efficiency and Scalability (Weakness 3)** Thank you for this suggestion. Overall, the offline memory construction and online memory querying are both highly efficient, largely due to the parallelization leveraged in our implementation. We have also collected some concrete efficiency metrics and will add them to the final version:
> - **Memory Construction (Offline)**: We re-ran the construction pipeline on SymPy, the repository with the largest history in the benchmark (>60K commits). The total end-to-end wall-clock time was approximately **45 minutes** (37 minutes for data crawling and 8 minutes for parallelized preprocessing and semantic memory construction). Note that this is also a one-time cost per repository.
> - **Inference Latency (Online)**: We re-ran RepoMem on a random 50-example subset of SWE-bench-Verified, recording the overhead of different components. The average total runtime per instance is **28.6 seconds**. Crucially, the overhead introduced by the memory search tools is almost negligible, averaging only **0.22 seconds** (relatedly, the original LocAgent tools also only cost 0.37 seconds). The vast majority of time is spent on LLM inference, confirming that RepoMem adds minimal online latency to the agentic loop.
>
> **4. Retrieval Generalization (Question 1)** We investigated whether the LLM-aware tokenizer generalizes beyond the Django repository. We compared vanilla BM25 against the LLM-based tokenizer across all repositories. The results below (Acc@5) confirm that the LLM-based tokenizer almost always outperforms the standard tokenizer across the board:
>
> | Repo | Matplotlib | Sympy | Astropy | Django | Sklearn | Sphinx | Pytest | Others |
> | :--- | :--- | :--- | :--- | :--- | :--- | :--- | :--- | :--- |
> | **BM25 (Standard)** | 79.4 | 69.9 | 81.8 | 77.9 | 90.6 | 59.1 | 72.2 | 47.8 |
> | **BM25 (LLM-based)** | **82.4** | **72.6** | **86.4** | **79.7** | **96.9** | **59.1** | **77.8** | **54.3** |
>
> We thank the reviewer again for the thoughtful comments and constructive suggestions. We will incorporate the improvements and additional results into the revised draft over the next few days. We look forward to addressing any additional questions or comments.

---

> > ### Author Response · Authors · 2025-11-24
> > **Help read the rebuttal and provide feedback**
> >
> > Dear Reviewer Fbe2, thanks again for your recognition of our work. We would greatly appreciate it if you could review our rebuttal and share your feedback. Please let us know if further clarification or additional responses are needed.

---

### Official Review · Reviewer_F6Cy · 2025-11-01

**Soundness:** 3
**Presentation:** 4
**Contribution:** 3
**Rating:** 8
**Confidence:** 5

**Summary:**

The authors propose integrating commit history and recent activity via a "memory" interface as context for LLM agents when tackling fault localisation. To demonstrate the value of their approach, they integrate with LocAgent and expose the memory through tools/API calls. They evaluate on SWE-bench-verified and SWE-bench-live, showing consistent improvement in performance over the vanilla LocAgent. The authors also analyse the tools employed by the agent during localisation and the cost trade-off depending on success (either of the memory-based tool or the vanilla tool), demonstrating that the cost landscape is non-trivial. This work serves as a first step in considering external project information sources that can serve as project/institutional memory (commit history, issue trackers, code reviews, etc.).

**Strengths:**

- combining project history (commit history) with repository structure for better localisation in the context of LLMs.
- demonstrated consistent improvement in performance over the extended baseline/SOTA
- analysis of the cost trade-off between using/not using memory tools
- careful removal of information that can cause data leakage (through issues/transitive relationships)

**Weaknesses:**

- The cost overhead is significantly higher, specifically for cases where vanilla LocAgent fails. (I am not convinced about tool allocation, especially since hotspots and recent development can be a very strong signal, at least in just-in-time defect prediction)
- The memory construction hyperparameters seem odd (I expected runtime compression + last 14 days of commits or similar thresholds from SE, not 7000 commits as pre-processing)
- Some of the metadata exposed/used by the proposed approach depends on links that are known to be noisy (Issue-Commit links are known to be missing: seminal paper for the problem definition -- Bachmann, Adrian, et al. "The missing links: bugs and bug-fix commits." Proceedings of the eighteenth ACM SIGSOFT international symposium on Foundations of software engineering. 2010. But research has progressed since)
- Memory construction may be inefficient, but the efficiency of memory construction may be out of scope. (reference paper -- S. Kim, T. Zimmermann, K. Pan and E. J. Jr. Whitehead, "Automatic Identification of Bug-Introducing Changes," 21st IEEE/ACM International Conference on Automated Software Engineering (ASE'06), Tokyo, Japan, 2006, pp. 81-90, doi: 10.1109/ASE.2006.23. And the area of just-in-time defect prediction). Specifically, borrowing ideas of how to narrow which commits to summarise/choose.

**Questions:**

- Comment1: While not directly related, a "sister" area of research is just-in-time defect prediction, and the literature reference for this is missing. Consider this survey as a starting point: Yunhua Zhao, Kostadin Damevski, and Hui Chen. 2023. A Systematic Survey of Just-in-Time Software Defect Prediction. ACM Comput. Surv. 55, 10, Article 201 (October 2023), 35 pages. https://doi.org/10.1145/3567550. The core of the comment is that, by the nature of the just-in-time version of the fault localisation problem, the data that is to be used is the commit history. Still, to my knowledge, this work is the first to consider this source of information in an agentic framework, enabling access through tools.

- Q1: In JIT defect prediction, features such as recent commits, recent commits by the same author, historically co-chaged files, etc., are used as manually created features. In this work, such metadata enrichment is not considered. Was such, or different, preprocessing considered beyond the scope of the work? For example, I would expect even a time-stamp filter to aid when retrieving development history for an agent to use; 7000 commits before the current base commit is considerably more than what is standard in SE work.

---

> ### Author Response · Authors · 2025-11-21
>
> We thank the reviewer for the highly positive assessment and for identifying our work as a solid first step in integrating long-term project memory into agentic frameworks. We particularly appreciate the expert connections drawn to Software Engineering (SE) literature, specifically Just-in-Time (JIT) defect prediction. We address your specific comments and questions below.
>
> **1. Cost Overhead**
>
> Indeed, cost efficiency is a critical consideration, for which we also dedicated substantial analysis. As our instance-level cost analysis (Figure 4 and lines 422-433) reveals, the cost increase is actually not uniform and exhibits high variance:
> - **Cost Savings**: In many cases, RepoMem drastically *reduces* costs because retrieving a similar historical patch or a relevant file summary leads the agent quickly to the error source, bypassing expensive brute-force codebase traversal.
> - **Cost Investments**: The average cost increase is largely driven by “novel” issue instances where the memory retrieval does not yield meaningful information. In these cases, the agent currently appears to try many different ways to query the memory, which leads to a large cost increase.
>
> We view this as a limitation of the current *prompting* strategy rather than our framework/approach itself. Since we use an off-the-shelf LLM, it lacks the intrinsic intuition to know *when* memory will be useful or *how* best to query the memory. For an experienced developer, he/she could have good ideas about the degree of “novelty” of a certain issue, and fall back to first-principle reasoning when the problem is very new. We believe a promising future direction—inspired by our analysis and your feedback—is to train agents to use these memory tools more strategically rather than exhaustively.
>
> **2. Memory Construction: Hyperparameters, Metadata & Overhead**
>
> You are entirely correct that 7000 commits is a heuristic derived from a standard hyperparameter search, rather than a domain-informed choice. We acknowledge that the design space for memory construction is vast.
> - **Future Optimization**: We fully agree that leveraging SE-specific heuristics—such as time-based windows, author filtering, or co-change history as used in JIT defect prediction—would likely result in a more efficient and higher-quality memory than our current fixed-window approach.
> - **Scope**: As this is the initial work introducing repository memory to agentic frameworks, we opted for a straightforward selection strategy to establish the efficacy of our approach. We will explicitly discuss these alternative, metadata-rich preprocessing strategies in the updated version as the logical next steps for improving our approach.
>
> In terms of overhead, our memory construction process is actually quite efficient, due to the high degree of parallelization in data fetching and preprocessing that we implemented. To make things concrete, we re-ran the memory construction pipeline on the repository with the largest number of commits in SWE-bench-verified—SymPy, which has over 60K historical commits. The end-to-end wall-clock time was only about 45 minutes: roughly 37 minutes for data crawling (commits, issues, etc.) and about 8 minutes for preprocessing and constructing the semantic memory of “hotspot” files (which is also parallelized). The overhead is largely a matter of engineering efforts; at even larger scales, we believe substantial optimization is possible as most of the computations here are parallelizable.
>
> **3. Noisy Metadata and Issue-Commit Links**
>
> Thank you for pointing out the Bachmann et al. (2010) reference. Throughout our project, we indeed encountered various noise sources, such as commits referencing issues using non-standard keywords that our rule-based parser missed. Despite these, our results demonstrate that the signal provided by even an imperfect memory is sufficiently strong to improve the agent’s performance. We will cite the suggested literature and discuss the different noises and opportunities to further improve the quality of the memory.
>
> **4. Connection to JIT Defect Prediction**
>
> We thank the reviewer for the excellent connection to JIT defect prediction. We have reviewed the suggested survey, and there is no doubt that it is a "sister" area that shares our basic intuitions. We will incorporate and discuss related references in our Related Work section to better ground our approach within the broader SE literature.
>
> We thank the reviewer again for the thoughtful comments and constructive suggestions. We will incorporate the improvements and additional results into the revised draft over the next few days. We look forward to addressing any additional questions or comments.

---

> > ### Author Response · Authors · 2025-11-24
> > **Help read the rebuttal and provide feedback**
> >
> > Dear Reviewer F6Cy, thanks for your recognition of our work. We would greatly appreciate it if you could review our rebuttal and share your feedback. Please let us know if further clarification or additional responses are needed.

---

> > > ### Comment · Reviewer_F6Cy · 2025-11-25
> > >
> > > Thank you for the answers, and they mostly align with my expectations that addressing more than just acknowledging the issues is beyond the scope of the paper. The wall-clock cost does move it from dev-interactive to CI/CD integration, but again, this is in the domain of applying/deploying and more a question I would have for an ICSE submission rather than ICLR.
> > >
> > > I maintain my positive score and do not have follow-up questions that would be in scope for ICLR.

---

### Author Response · Authors · 2025-11-26
**Manuscript Revision**

We thank all reviewers for the thoughtful and constructive feedback. We have carefully revised the manuscript to address the concerns raised and incorporate the suggested improvements. The major updates include:
- **Memory Construction and Overhead**: We added Appendix A to provide more details about the memory construction, as well as the offline overhead and online latency (**Reviewers F6Cy, Fbe2, tvHt, and bN9m**).
- **Additional Experimental Results (Appendix B)**: We have included additional experimental results, including sensitivity analysis for semantic memory construction (**Reviewer tvHt**), impact of localization on issue resolution rate (**Reviewer bN9m**), and retrieval generalization analysis (**Reviewer Fbe2**).
- **Expanded Related Work**: We added Section 2.2 to discuss relevant Software Engineering literature, specifically establishing connections to Just-in-Time (JIT) defect prediction (**Reviewer F6Cy**).

We believe these revisions strengthen the paper’s empirical rigor and its grounding in broader SE literature. We look forward to addressing any further questions or comments.

---

### Meta-Review · Area_Chair_wFPV · 2026-01-05

**Summary:**

This paper addresses the lack of long-term repository memory in language agents for code localization. Inspired by human developers, it leverages commit history to build episodic memory from past commits and issues, and semantic memory summarizing actively evolving code. Integrated into LocAgent, this memory-augmented approach significantly improves localization performance on SWE-bench-verified and SWE-bench-live. The results demonstrate that incorporating experience-driven memory enables more effective, human-like reasoning in repository-level software engineering tasks.

**Reviewer Concerns:**

The reviewers raise concerns regarding cost overhead, memory construction efficiency, and reliance on noisy metadata such as issue–commit links. The heuristics for identifying “active code” lack statistical justification and may not generalize beyond SWE-bench variants, with performance degrading on repositories with sparse histories. Design choices around large-scale preprocessed commit memory (e.g., 7k commits) and top-k file summaries appear ad hoc. The approach offers limited conceptual novelty and lacks end-to-end SWE-bench impact, scalability, and sensitivity analyses.

After the rebuttal, Reviewer F6Cy and Reviewer tvHt think their concerns have been solved, and Reviewer tvHt has raised his score from 4 to 6. As for other reviewers, I think most concerns would be addressed.

**Reviewer Scores:**

Reviewer F6Cy has replied and keeps 8 unengaged
Reviewer tvHt has replied and raised it to 6
I think the Other two reviewers would keep their score unchanged, 6 and 4

---

### Decision · Program_Chairs · 2026-01-26

Accept (Poster)